# Polymorphism and Conformational Equilibrium of Nitro-Acetophenone in Solid State and under Matrix Conditions

**DOI:** 10.3390/molecules26113109

**Published:** 2021-05-22

**Authors:** Łukasz Hetmańczyk, Przemysław Szklarz, Agnieszka Kwocz, Maria Wierzejewska, Magdalena Pagacz-Kostrzewa, Mikhail Ya. Melnikov, Peter M. Tolstoy, Aleksander Filarowski

**Affiliations:** 1Faculty of Chemistry, Jagiellonian University, Gronostajowa 2, 30-387 Kraków, Poland; hetmancz@chemia.uj.edu.pl; 2Faculty of Chemistry, Wrocław University, I14 F. Joliot-Curie st., 50-383 Wrocław, Poland; przemyslaw.szklarz@chem.uni.wroc.pl (P.S.); agnieszka.kwocz@chem.uni.wroc.pl (A.K.); maria.wierzejewska@chem.uni.wroc.pl (M.W.); magdalena.pagacz-kostrzewa@chem.uni.wroc.pl (M.P.-K.); 3Department of Chemistry, Moscow State University, F. Joliot-Curie 14, 119991 Moscow, Russia; melnikov46@mail.ru; 4Institute of Chemistry, St. Petersburg State University, Universitetskij pr. 26, 198504 St. Petersburg, Russia; peter.tolstoy@spbu.ru; 5Frank Laboratory of Neutron Physics, Joint Institute of Nuclear Research, 141980 Dubna, Russia

**Keywords:** polymorphism, isomerization, phase transition, nitro group, matrix isolation, IINS, FT-IR, Raman, X-ray, NQR, DSC, DFT

## Abstract

Conformational and polymorphic states in the nitro-derivative of *o*-hydroxy acetophenone have been studied by experimental and theoretical methods. The potential energy curves for the rotation of the nitro group and isomerization of the hydroxyl group have been calculated by density functional theory (DFT) to estimate the barriers of the conformational changes. Two polymorphic forms of the studied compound were obtained by the slow and fast evaporation of polar and non-polar solutions, respectively. Both of the polymorphs were investigated by Infrared-Red (IR) and Raman spectroscopy, Incoherent Inelastic Neutron Scattering (IINS), X-ray diffraction, nuclear quadrupole resonance spectroscopy (NQR), differential scanning calorimetry (DSC) and density functional theory (DFT) methods. In one of the polymorphs, the existence of a phase transition was shown. The position of the nitro group and its impact on the crystal cell of the studied compound were analyzed. The conformational equilibrium determined by the reorientation of the hydroxyl group was observed under argon matrix isolation. An analysis of vibrational spectra was achieved for the interpretation of conformational equilibrium. The infrared spectra were measured in a wide temperature range to reveal the spectral bands that were the most sensitive to the phase transition and conformational equilibrium. The results showed the interrelations between intramolecular processes and macroscopic phenomena in the studied compound.

## 1. Introduction

The aim of this paper is to investigate polymorphic and conformational states of the nitro-derivative of *o*-hydroxy acetophenone. The study explains the influence of the intramolecular hydrogen bond on the phase transition and conformational equilibrium. The expected intramolecular dynamic processes in the *o*-hydroxy acetophenone molecule are the rotation of the nitro group and isomerization of the hydroxyl group (Scheme 1). These processes define such macroscopic phenomena as polymorphism, phase transition and the existence of stable isomers under different conditions. The study of conformational isomerism is a very important issue in modern chemistry for determining and modelling the physical–chemical properties of new materials [1,2,3,4] and pharmaceutical compounds [5,6,7,8]. It must be stressed that an intramolecular hydrogen bond strongly influences the isomerization equilibrium [9,10]. For deeper insight into the hydrogen-bonding effect on the polymorphic states and isomerization we applied a wide variety of research methods (DFT, X-ray, DSC, NQR, IINS, IR and Raman) in different environments and over a wide temperature range. It is worth noting that the method of matrix isolation was unique in its ability to trace metastable states and help interpret complex reactions [11,12,13,14]. Moreover, investigations of objects with hydrogen bonding by IINS [15,16,17,18] and NQR [19,20] techniques are useful.

In this paper the research methodology followed the following sequence: First, DFT calculations were performed to detect stable and metastable states of the studied molecule, 5-chloro-3-nitro-2-hydroxyacetophenone (**CNK**). The first stage of the studies predicted that **CNK** would crystallize into two polymorphic forms; therefore, its structural properties were analyzed by X-ray diffraction at different temperatures, and both polymorphs were studied by the DSC method to detect phase transition, which was found in one of the polymorphs and verified by ^35^ Cl NQR measurement. Spectral properties of both polymorphs were investigated by IINS, IR and Raman, and computational (DFT) methods in different states to obtain exhaustive information about which vibrational bands (as well as their assignments) were the most sensitive to the phase transition. The assignments of the spectral bands were accomplished on the basis of the H/D substitution of bridge hydrogen (OH → OD) and Potential Energy Distribution (PED) analysis. Spectral infrared studies of the two isotopologues (OH and OD) showed the presence of two conformers (A and B, Scheme 1) under the argon matrix condition.

## 2. Results and Discussion

At first, the studies dwelt on the DFT calculations of the potential curves to detect stable and metastable states of the molecule according to the concepts presented in the review paper by Bernstein [2]. The information about the energy difference between the global and local minima as well as the height of energy barriers made it possible to predict the presence of a particular conformer or polymorph depending on the environment. For the studied molecule it was logical to consider two possible intramolecular processes—reorientation of the hydroxyl group and rotation of the nitro group (Scheme 1). Therefore, the dependencies of the potential energy on the turning angle of the hydroxyl and acetyl groups were obtained by DFT calculations, which were performed under step-by-step changes to the torsional angle while all other geometric parameters of the molecule were optimized. The calculation of potential energy dependence on the Θ angle of turning the nitro group is given by the equation Δ*E* = *f*(Θ), where Δ*E* = *E*_min_ − *E*_i_; *E*_min_ is the minimal energy of the system; and *E*_i_ is the energy of the system for each fixed Θ angle). The result demonstrated that the energy barrier was rather small and equaled 0.78 kcal/mol when the Θ angle equaled 0 degrees (Figure 1).

Such a result may seem unconvincing because the torsional angle at 0 degrees (a flat molecule) is usually characterized by minimal potential energy because of π-electronic coupling between the nitro group and phenyl moieties. However, the studied molecule displayed a significant steric repulsion between the oxygen of the nitro group and the oxygen of the hydroxyl group, which counteracts the π-electronic coupling. Such steric repulsion results in the appearance of an energy barrier (Δ*E* = 0.78 kcal/moL) at Θ = 0° (Figure 1). Notably, this barrier is not large; thus, the nitro group can easily change its position with respect to the phenyl ring. The process of reorientation of the hydroxyl group leading to the transition from conformer A to conformer B is more complicated and is presented in Appendix A. Judging from the calculated energy barrier of 14–15 kcal/mol, one expected the existence of both conformational forms at a reasonable temperature [21]. Taking these findings into account, we assumed that the studied compound featured polymorphism brought about by the rotation of the nitro group, or by a significant conformational change introduced by the reorientation of the hydroxyl group.

We performed comprehensive studies to trace polymorphs, phase transitions and conformational change. Two polymorphs of the **CNK** compound were obtained by slow recrystallization from methanol (polymorph **I**) and fast re-crystallization from chloroform (polymorph **II**). X-ray studies of polymorphs **I** and **II** showed that they crystallized in a Pccn (T = 200 K) and P21/c (100 K) space group, respectively. The comparison of the structures of the polymorphs clearly showed the difference in the position of the nitro group (Figure 2 and Appendix A). The disoriented position of this group in polymorph **I** points to significant dynamics in the solid state. The crystal cell of polymorph **II** is characterized by a more defined orientation of the nitro group. The nitro groups of polymorph **I** are able to rotate, but the nitro groups of polymorph **II** are not since in the crystal cell of polymorph **II** the molecules are packed in a way that turns the nitro groups and blocks the rotation. The studies of both polymorphs were carried out by DSC to detect a phase transition. The DSC measurements of both polymorphs showed the presence of a phase transition for polymorph **I** at 109.8 K (cooling)/114.5 K (heating) (Appendix A) and the absence of a phase transition for polymorph **II**. The phase transition is reversible and the transformation is enantiotropic.

The comparison of the structural data and the packing of the molecules in the crystal cell of both polymorphs made it possible to conclude that the phase transition was conditioned by the following phenomenon. A decrease in temperature tightened the packing of the molecules in the cell and, therefore, caused a stronger interaction between the nitro groups. Polymorph **II** featured “jagged” nitro groups, which evoked the stable position of molecules in the crystal cell upon temperature decrease. Polymorph **I** lacked this phenomenon, having the possibility of a looser nitro-group rotation at room temperature, which stopped when the temperature was lowered. The decreasing temperature led to stronger interactions between nitro groups and shifted the molecules towards each other, triggering the phase transition. Such a conclusion was verified by comparing the packaging of **CNK** to that of the structurally similar 5-methyl-3-nitro-2-hydroxyacetophenone (**MNK**) [22]. For **MNK,** the nitro groups were oppositely directed and not able to interact strongly, so they did not provoke tensions in the crystal cell. Therefore, the structurally close **MNK** did not exhibit phase transition and polymorphism (cf. Appendix A).

The phase transition in **CNK** was detected on the basis of NQR measurements. NQR was successfully used in the research of compounds with intermolecular [23] and intramolecular hydrogen bonds [24]. NQR studies revealed that the ^35^ Cl signal shifted to high frequencies (from 35.85 to 36.5 MHz) when the temperature fell from 300 to 120 K (the temperature of phase transition), whereas the signal was quite stable at temperatures below the transition state (Figure 3). A similar trend was observed for 1,3-diazinium hydrogen chloranilate monohydrate [25], morpholinium hydrogen chloranilate [26] and *p*-dichlorobenzene [27].

After comparing the structural data and the crystal packing of both polymorphs we concluded that the polymorphism and phase transition were conditioned by the position of the nitro group. Based on experimental data, this conclusion was in accordance with theoretical predictions.

### 2.1. Detection of Polymorphs in the Solid State and Isomers under the Matrix Condition by Spectroscopic Methods

To detect the spectral bands that are the most sensitive to polymorphic and conformational changes as well as to phase transition, we performed an analysis of the vibrational spectra measured in solid state and under the matrix condition. To that end, IR, Raman and IINS spectra of the studied compound and its deuterated derivative (OH → OD) were measured in the wide spectral and temperature ranges (50–4000 cm^−1^, 300–5 K, Figure 4 and Figure 5 and Appendix A). The analysis of the spectra and the assignments of the bands were based on DFT and PED calculations (Appendix A). Below, the description of the spectra measured in the solid state and the matrix isolation condition is presented, on the basis of which the isomeric equilibrium analysis was carried out.

#### 2.1.1. The ν(OH) Stretching Mode

The adequate assignments of the bands of stretching (ν(OH)), in-plane (δ(OH)) and out-of-plane (γ(OH)) bending modes of the hydroxyl group, as well as the isotope effects of these vibrations caused by the replacement of the bridged hydrogen by deuterium (OH → OD) was completed.

*Solid state.* When comparing the infrared spectra of deutero-(**CNK-OD**) and non-deutero-(**CNK**) derivatives of the studied compound, the band at 3000–2100 cm^−1^ in the IR spectra recorded in the solid state was assigned to the ν(OH) stretching mode due to its shift to 2100–2000 cm^−1^ after deuteration (Appendix A). It was necessary to underline that the measured infrared spectra in the solid state did not reveal the intense band at 3100 cm^−1^ that was assigned to the stretching vibration of the hydroxyl group, which is hydrogen bonded with the nitro group (cf. spectra of **CNK** and *o*-nitro-phenol [28]). This fact confirmed the absence of conformational form **A** of the studied compound in the solid state. The result agreed with the presented X-ray study.

*Matrix condition.* The analysis of the ν(OH) band in the infrared spectrum measured under the matrix conditions does not provide a clear proof for the presence of two conformers. The reason is the overlapping of ν(OH) bands of both conformers and ν(CH) bands. However, two bands at 2185 cm^−1^ and 2350 cm^−1^ appear in the spectrum of **CNK-OD**; they are assigned to the ν(OD) vibration of conformers **A** and **B** (Figure 4), respectively. This result manifests the presence two conformational forms under the matrix condition.

#### 2.1.2. The δ(OH) and γ(OH) Bending Modes

*Solid state.* The in-plane bending mode (δ(OH)) was hard to analyze because of an uncharacteristic vibration in the solid state. According to PED analysis of the spectra calculated by the DFT method, this vibration was strongly coupled to the stretching vibrations of the phenyl ring (1563 and 1433 cm^−1^, Appendix A).

A broad band at 860 cm^−1^ for polymorph **I** and the band at 838 cm^−1^ for polymorph **II** (Appendix A) were assigned to the out-of-plane mode of the hydroxyl group (γ(OH)) in IR spectra measured in the solid state. The bands at 860 cm^−1^ and 838 cm^−1^ narrowed down drastically under the deutero replacement, and new bands arose at 628 cm^−1^ and 624 cm^−1^ in the **CNK-OD** spectra (Appendix A, ESI). These bands were assigned to the γ(OD) mode according to ISR = 1.35 (ISR—isotopic spectroscopic ratio).

*Matrix condition.* As for in-plane and out-of-plane bending modes, a more distinct picture (due to the absence of the overlapping bands) was observed in IR spectra measured under the matrix condition. In these spectra, obvious changes to two bands at 1269 cm^−1^ and 1166 cm^−1^ appeared after deuteration. They completely disappeared after deuteration and new bands appeared at 959 cm^−1^ and 895 cm^−1^ in the **CNK-OD** spectrum (Figure 4). According to the observed changes and the calculated isotopic ratio (ISR = 1.32), these two bands were assigned to the δ(OH) and δ(OD) modes of conformer **B** and conformer **A**, respectively (Appendix A). Rather broad and weak bands at 823 cm^−1^ and 719 cm^−1^ completely vanished and turn up at 619 cm^−1^ and 534 cm^−1^ after deuteration (ISR = 1.33), the bands having been assigned to the γ(OH) and γ(OD) modes of conformer **B** and **A**, respectively. The results presented above are in agreement with the data based on PED analysis (Appendix A) as well as the data obtained earlier for 5-methyl-3-nitro-2-hydroxyacetophenone [29].

#### 2.1.3. The ν(C = O) Stretching Mode

It is noteworthy that the stretching vibration of the carbonyl group (ν (C = O)) was the most sensitive to the conformational equilibrium [30,31]. According to IR and Raman spectra measured in the solid state (Appendix A), only one band was observed in the 1800–1600 cm^−1^ range at 1650 cm^−1^ and was assigned to ν (C = O) mode (PED analysis, Appendix A). Interestingly, the ν (C = O) bands of both polymorphs were nearly the same, and therefore, demonstrated very little sensitivity to polymorphic changes. However, IR spectra under the matrix condition were characterized by two bands at 1700 cm^−1^ and 1667 cm^−1^, which were assigned to the ν (C = O) vibrations of conformers **A** and **B** (Figure 4), correspondingly. This statement is in accordance with our previous studies of the methyl derivative of *o*-hydroxy acetophenone [29].

#### 2.1.4. Nitro Group Mode

*Solid state*. The bands most sensitive to polymorphic states were assigned to the nitro group vibrations. The bands at 1533 cm^−1^ (ν^as^ (NO_2_)), 1346 cm^−1^ (ν^sym^ (NO_2_)), 900 cm^−1^ (δ(NO_2_)) in the Raman spectrum and at 899 cm^−1^ (δ(NO_2_) in the IR spectrum shifted noticeably upon transitioning from polymorph **I** to polymorph **II** (1540 cm^−1^, 1357 cm^−1^, 881 cm^−1^ in Raman and 894 cm^−1^ in IR spectra, Appendix A). A large increase in intensity of the band at 1346 cm^−1^ was also detected in the Raman spectrum upon transition from polymorph **I** to polymorph **II**.

*Matrix condition.* Two intense bands at 1550 cm^−1^ and 1539 cm^−1^ as well as two bands at 1356 cm^−1^ and 1303 cm^−1^ observed in IR spectrum registered under the matrix condition (Figure 4) were assigned to asymmetric and symmetric vibrations of the nitro group of conformers **B** and **A**, respectively. As for the solid state, the bands assigned to conformer **B** (cf. IR spectra obtained under the matrix condition and in the solid state, Figure 4 and Appendix A) were absent in IR and Raman spectra.

### 2.2. Spectral Changes under Phase Transition in the Solid State

Regarding phase transition, no significant band splitting during the transition from one phase to the other was observed in IR spectra measured in the middle spectral range. However, the IR spectrum in the far-infrared range demonstrated visible changes (Appendix A). At higher temperatures (300–120 K) there were single bands at 454, 409, 350 and 182 cm^−1^, which split into doublets at the temperature below the phase transition. The IR spectra of polymorph **II** were measured to verify the results, which did not reveal the splitting of the abovementioned bands in the far-infrared range for the polymorph without the phase transition. The PED analysis showed that bands at 454, 350 and 182 cm^−1^ belonged to multicomponent modes (Appendix A). The intense band at 409 cm^−1^ in the IR spectrum was assigned to ν_σ_ (OHO) vibration due to a strong reduction of the intensity upon deutero replacement. The reliability of the assignment was also proved by IINS studies (Figure 5).

For a deeper insight into the vibrational spectra, isotopic effect and conformational polymorphism we studied the IINS spectra of **CNK** and **CNK-OD** within the 1200–0 cm^−1^ range in the solid state (Figure 5). These spectra showed the presence of two polymorphic forms at low temperature and verified the correctness of the assignment of the vibrations of the hydroxyl group and the hydrogen bridge (ν_σ_ (OHO)).

Most bands of the measured IINS spectra of both isotopologues were doublets (Figure 5; in the figure, the doublets are marked by filled squares, circles and triangles). In the papers by Tomkinson [32] and Margues [33], it is stated that the split bands in the IINS spectra occurred as a result of two crystallographically and energetically non-equivalent modes. The doubling of bands in the IINS spectra was also supposed to result from the Davydov effect [34], which is the separation of energy levels ascribed to the same vibration due to the presence of several interacting molecular entities in the unit cell. Therefore, the splitting of the bands in the IINS spectra in both isotopologues was due to the presence of two polymorphic forms at 10 K. Three bands at 949, 916 and 895 cm^−1^ were observed within the range of 1000–850 cm^−1^, three of which were the result of th overlapping of two doublets: 949/916 cm^−1^ (circles) and 916/895 cm^−1^ (triangles). The doublet at 895/916 disappeared upon deuteration (cf. spectra **CNK** and **CNK-OD**, Figure 5) and appeared as a very small doublet at 656 and 676 cm^−1^. According to the calculated ISR coefficient (ISR = 1.35) and the PED analysis (Appendix A), these bands were assigned to the γ(OH) and γ(OD) out-of-plane bending modes. The values of the γ(OH) bands of IR and IINS spectra were compared in a qualitative way with the values obtained from the correlation R(OO) = 3.01 + 0.0044 × 10^−4^ γ(OH) (for OHO hydrogen bridges longer than 2.4 Å) presented in reference [35]. The position of the γ(OH) band shifted towards the high frequencies alongside the strengthening of the hydrogen bond (823 cm^−1^ in matrix condition <860 cm^−1^ in the solid state). This fact supports the tendency shown in the review by Novak [36] for medium-strong intramolecular hydrogen bonding.

## 3. Materials and Methods

### 3.1. Chemicals

Compounds and solvents were purchased from Sigma-Aldrich and used without purification. High-purity argon gas (N60 = 99.99990%) was obtained from Air Liquide. The deuterated sample was prepared by dissolving the product in deuterated methanol (CH_3_OD). The solution was heated to 60 °C and refluxed for 20 min. Then the methanol was removed by evaporation under reduced pressure. This procedure was repeated three times. The deuteration degree was estimated to be ca. 80–90%.

### 3.2. Computational Details

Quantum-mechanical calculations using the B3LYP functional [37,38] with the 6-311++G(2d,2p) [39,40] basis set were performed with the GAUSSIAN 16 program [41]. The non-adiabatic approach was used to calculate Δ*E* = *f*(d(Θ)) dependence. Structural parameters were optimized for each fixed angle, changing gradually by 10°. The potential energy distribution (PED) of the normal modes for each obtained equilibrium geometry was calculated using the internal coordinates by the GAR2PED program [42].

### 3.3. Thermal Measurements

Differential scanning calorimetry (DSC) experiments were recorded with PerkinElmer Model 8500 differential scanning calorimeter on the polycrystalline material in the temperature range of 100–300 K under a nitrogen atmosphere in hermetically sealed Al pans. Calibration was performed with *n*-heptane and indium as standards.

### 3.4. NQR Measurements

The powdered sample without further preparation was placed in a standard 5 mm glass sample tube, which was subsequently placed inside the probe of the NQR spectrometer. ^35^ Cl NQR spectra were recorded using NMR/NQR Tecmaq Redstone spectrometer by the single pulse method in the frequency range 35–37 MHz. The pulse duration was 3 µs, followed by 1 ms acquisition and subsequent 400 ms delay time. The number of scans was ca. 3000. The temperature was controlled using an Oxford Instrument cryostat and stabilized with the precision better than ±1 K.

### 3.5. Crystallographic Details

The intensity data were collected at 100 K using a Kuma KM4CCD diffractometer and graphite-monochromated MoKα (0.71073 Å) radiation generated from an X-ray tube operating at 50 kV and 35 mA. The images were indexed, integrated, and scaled using the Oxford Diffraction data reduction package [43]. The experimental details together with crystallographic data are given in Appendix A. An absorption correction was omitted. The structure was solved by direct methods using SHELXS97 [44] and refined by the full-matrix least-squares method on all F^2^ data (SHELXL97) [45]. Non-hydrogen atoms were refined with anisotropic thermal parameters; hydrogen atoms were included from Δ*ρ* maps and refined isotropically. The supplementary crystallographic data for this paper is deposited to Cambridge Crystallographic Data Centre (CCDC) under no. 1937047-1937049 for 5-chloro-3-nitro-2-hydroxyacetophenone. These data can be obtained free of charge via www.ccdc.cam.ac.uk/conts/retrieving.html (accessed on 11 May 2021) (or from the Cambridge Crystallographic Data Centre, 12 Union Road, Cambridge CB2 IEZ, UK: fax: (+44) 1123-336-033. e-mail: deposit@ccdc.cam.ac.uk). The molecular structures with atom labeling are shown in Appendix A.

### 3.6. IR and Raman Measurements

The standard infrared and Raman spectra were measured using Bruker IFS 66 FT-IR and Nicolet Magna 860 FT-Raman spectrometers in the solid state with a 2 cm^−1^ resolution, respectively. The In–Ga–Ar laser line at 1064 nm was employed for the Raman excitation measurements. To obtain matrices containing **CNK**, the crystalline sample was allowed to sublimate from a small electric oven located inside the vacuum vessel of the cryostat. The **CNK** vapours, mixed with a large excess of matrix gas (argon), were deposited onto a CsI window kept at 15 K in a closed cycle helium cryostat (APD-Cryogenics). The sample temperature was maintained with a temperature controller (Scientific Instruments 9700) equipped with a silicone diode and a resistive heater. Infrared spectra were recorded at 11 K between 4000 and 50 cm^−1^ with a resolution of 0.5 cm^−1^ by means of a Fourier transform IR spectrometer (Bruker IFS 66) equipped with a liquid N_2_ cooled MCT detector.

### 3.7. Incoherent Inelastic Neutron Scattering (IINS) Measurement

Neutron scattering data were collected at the pulsed IBR–2M reactor at the Joint Institute of Nuclear Research (Dubna, Russia) using the time-of-flight inverted geometry spectrometer NERA-PR at 10 K. To avoid misunderstanding in the comparison of the IINS spectra, the samples of the studied compounds were measured as a powered substance.

## 4. Conclusions

To summarize, the paper showed an application of quantum-mechanical calculations for the detection of particular physical-chemical processes in a molecule. DFT calculations enabled the prediction of the possible observation of two phenomena for the studied compound: polymorphism in the solid state and the conformational equilibrium under the condition approaching to the gas phase. The first of the predicted phenomena—the existence of two polymorphs of the studied compound and the phase transition between them-was revealed by DSC and NQR techniques. The non-equivalency of the structures in crystal cells was reflected in the splitting of the bands in the IINS spectrum, and the bands in IR spectrum registered in far-infrared region.

The second of the predicted phenomena was the conformational equilibrium between two hydrogen bonds (N = O⋯H–O and O–H⋯O = C(CH_3_)). Two conformational forms detected in the matrix isolation conditions by infrared spectroscopy unambiguously proved this prediction. Complete assignments of spectral bands based on the PED analysis and isotopic substitution showed the presence of two conformers under the matrix condition and two polymorph forms in the solid state. The studies confirmed the agreement of the theoretical and experimental results.

## Data Availability

All data associated with this article are included in Appendix A.

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
