# Peer review of "Polymorphism and Conformational Equilibrium of Nitro-Acetophenone in Solid State and under Matrix Conditions"

_molecules, 2021, doi:10.3390/molecules26113109_

Round 1
Reviewer 1 Report
This MS considers the polymorphism of the nitro-derivative of o-hydroxy acetophenone in the solid state and its conformational equilibrium in near gas phase state. The multitemperature Infra-Red and Raman spectroscopy, Incoherent Inelastic Neutron Scattering, single-crystal X-ray diffraction, Nuclear Quadrupole Resonance, Differential Scanning Calorimetry and DFT calculations were applied to explain the influence of H-bond on these phenomena. First, the DFT calculations were used to detect stable and metastable states of the studied compound; then, the experimental characterization has been done.
The results are :
Two polymorphs of the studied compound have been obtained in the solid state. The phase transition in one of the polymorphs has been revealed by DSC and NQR and the nonequivalency of the structures in a crystal, reflected in the splitting of the bands in IINS spectrum and the bands in IR spectrum was found in far-IR region, was established.
Two conformational forms were detected in the matrix isolation by IR spectroscopy: it supports the prediction that the conformational equilibrium
between two hydrogen bonds N=O...H–O and O–H...O=C(CH3) takes place.
In general, the MS presents a nice piece of work showing the importance of the smart combination of modelling and measurements.
I have no serious critical comments. However, the English must be carefully polished.
Author Response
Dear Editors, Prof. M. Jablonski and Ms. Lucy Chai
We are grateful to the Reviewers for the deep reviews and the editors for the professional work.
We have introduced changes following the remarks made by the Reviewers.
Reviewer 1.
We thoroughly checked the text and corrected stylistic mistakes. The corrections are marked green in the text.
Sincerely,
Aleksander Filarowski
Reviewer 2 Report
The presented work was performed using a variety of experimental methods. The problem posed by the authors has been solved. Therefore, I recommend that you accept it for publication. At the same time, the work raises some questions.
- It is not clear how important this compound is and why it was necessary to spend large experimental efforts to study it.
- The authors used many different methods to solve the problem. However, the information received did not always have the required depth. For example, it is not clear why the expensive matrix isolation technique had to be used? To get an IR spectrum at one temperature point? Wouldn't it be better to get an accurate temperature dependence of the IR or Raman spectra and follow the same C-H or C = O vibrations in the region from helium to room temperature? It seems to me that in this case the authors would have received much more interesting and convincing results.
- The phrase about anharmonic vibrations (lines 174-176) is inappropriate and should be deleted.
Author Response
Dear Editors, Prof. M. Jablonski and Ms. Lucy Chai
Reviewer 2.
- A core value of the studied compound lies in two discovered physical phenomena. Fundamental studies have always been important and it is impossible to practically design new materials ignoring them. When it comes to the application, this type of compounds (o-hydroxy acetophenones) is widely used in cosmetics industry for UV radiation protection.
- We agree with the Reviewer opinion that more precise temperature IR and Raman studies will demonstrate new interesting results. We are planning to accomplish the proposed studies for other acetophenones in the future.
- The phrase about anharmonic vibrations (lines 174-176) is removed from the text.
Also we have a suggestion to the Editorial Board to consider our graphical abstract for the volume cover illustration.
Sincerely,
Aleksander Filarowski